# Supramolecular Hybrids from Cyanometallate Complexes and Diblock Copolypeptide Amphiphiles in Water

**DOI:** 10.3390/molecules27103262

**Published:** 2022-05-19

**Authors:** Takayuki Tanaka, Keita Kuroiwa

**Affiliations:** Department of Nanoscience, Faculty of Engineering, Sojo University, 4-22-1 Ikeda, Nishi-ku, Kumamoto 860-0082, Japan; takayukit12296@gmail.com

**Keywords:** self-assembly, metal complex, cyanometallate, nanostructure, photoluminescence, metal-metal interaction, nanorod, diblock copolypeptide, amphiphile

## Abstract

The self-assembly of discrete cyanometallates has attracted significant interest due to the potential of these materials to undergo soft metallophilic interactions as well as their optical properties. Diblock copolypeptide amphiphiles have also been investigated concerning their capacity for self-assembly into morphologies such as nanostructures. The present work combined these two concepts by examining supramolecular hybrids comprising cyanometallates with diblock copolypeptide amphiphiles in aqueous solutions. Discrete cyanometallates such as [Au(CN)_2_]^−^, [Ag(CN)_2_]^−^, and [Pt(CN)_4_]^2−^ dispersed at the molecular level in water cannot interact with each other at low concentrations. However, the results of this work demonstrate that the addition of diblock copolypeptide amphiphiles such as poly-(L-lysine)-*block*-(L-cysteine) (Lys_m_-*b*-Cys_n_) to solutions of these complexes induces the supramolecular assembly of the discrete cyanometallates, resulting in photoluminescence originating from multinuclear complexes with metal-metal interactions. Electron microscopy images confirmed the formation of nanostructures of several hundred nanometers in size that grew to form advanced nanoarchitectures, including those resembling the original nanostructures. This concept of combining diblock copolypeptide amphiphiles with discrete cyanometallates allows the design of flexible and functional supramolecular hybrid systems in water.

## 1. Introduction

The structure and function of naturally occurring supramolecular materials such as metalloproteins are determined by the arrangement of the primary, secondary, tertiary, and quaternary structures of polypeptide chains with various metal complexes [1,2,3,4,5,6,7,8,9,10,11]. In order to elucidate the complicated combinations of peptides with metal complexes, as well as the functions of these structures, biological polypeptides (that is, apoproteins), having complicated primary sequences, and their derivatives are typically used as models. However, the analysis of the conformations and configurations of synthetic amphiphilic peptides containing various hydrophilic and hydrophobic parts is also of interest [12,13,14,15,16,17,18,19,20,21]. Artificial diblock copolypeptide amphiphiles consisting of simple sequences of amino acids with hydrophilic and hydrophobic side chains have been reported to form diverse nanostructures based on variation in composition, despite their simple sequences. Such peptides have been found to exhibit micelle [22], fiber [23,24,25], tube [26,27,28,29,30,31], sheet [32,33], and capsule morphologies [34,35,36,37,38] based on the self-assembly of their hydrophilic and hydrophobic blocks. In a previous study of diblock copolypeptide amphiphiles, Deming et al. showed for the first time that these materials can be precisely synthesized from amino acid *N*-carboxyanhydrides (NCAs) by ring-opening living polymerization using organometallic initiators [39,40,41,42,43,44,45]. These synthetic peptides have many features that make them of interest to those working in the field of protein engineering, in applications such as drug delivery systems and tissue engineering.

Nanostructures of amphiphilic polypeptides in combination with metal complexes are able to be controlled for applications in biotechnology (as biosensors, artificial tissues, or implants) and biomineralization (as resilient, lightweight, and ordered inorganic composites) [46,47,48,49,50,51]. The combination of various transition metals with peptides can also affect the three-dimensional accumulation of metal ions. For example, the formation of polyhedral peptides using oligopeptide chains and metal ions and the use of these materials as artificial enzymes due to them having large internal cavities was recently reported [52,53,54]. These previous studies suggest that amphiphilic polypeptides can provide a suitable “bottom-up” approach to nanofabrication because the nanostructures of these polymers can be controlled by changing the ratio of hydrophobic to hydrophilic groups.

In our previous work, diblock copolypeptide amphiphiles were found to form flexible hybrid structures with metal complexes via a self-assembly process in water. These supramolecular hybrids possess specific nanostructures such as wires, ellipses, squares, and rectangles, hierarchically formed from lamellar and/or cylindrical structures in hybrids with copolypeptides [55,56,57,58]. The hybrid of polypeptides with metal complexes likely has some hierarchical formation, contributing at least partially to the enhanced incorporation of the self-assembling process. These nanostructures also showed reverse spin transitions [55], lower critical solution temperature (LCST)-type phase behavior with spin crossover phenomena [56], catalytic redox reaction based on the formation of two-dimensional metal-organic frameworks (2D-MOFs) [57], and photoluminescence [58], none of which had been previously observed in studies of these materials either in amorphous or crystalline forms. This prior research demonstrated that combining amphiphilic copolypeptides with metal complexes resulted in intermolecular interactions leading to weak self-assembly, dynamic transformations, and stimuli responsiveness in water. The inherent self-assembly abilities of these copolypeptide amphiphiles could potentially lead to their application as structural templates for inorganic compounds and to the intelligent transformation of inorganic materials involving dynamic tuning of electronic states.

The present study employs cyanometallate complexes as simple tools to produce supramolecular hybrids. It is known that *d*^10^ gold(I) or silver(I) and *d*^8^ platinum(II) complexes, in particular, will aggregate through *d*^8^-*d*^8^ or *d*^10^–*d*^10^ closed-shell metallophilic bonding interactions, and this process determines both the supramolecular structures and luminescent properties of these materials [59,60,61,62]. As a result of their dynamic luminescence properties, [Au(CN)_2_]^−^, [Ag(CN)_2_]^−^, and [Pt(CN)_4_]^2^^−^ all have applications as functional materials within intelligent molecular systems. Both the wavelength and intensity of the luminescent emissions of these complexes can be tuned based on aggregation through metal-metal bonding interactions, although high concentrations (>10 mM) are required for luminescence at ambient temperatures [59,60,61,62]. At present, the relationships between molecular structure, metal-metal interactions, and morphology at the nanoscale level are not thoroughly understood with regard to their influence on supramolecular structure. It is known that simple polypeptides and polymers can generate polyelectrolytes with [Au(CN)_2_]^−^ and promote the self-assembly of oligomeric and polymeric [Au(CN)_2_]^−^ structure [58,63]. However, it is unclear how synthetic polymers such as copolypeptide structures can be used to tune not only intermolecular interaction but also mesoscopic self-assembly. The ability to tune the nanoscale morphology of these materials via changes in molecular structure could potentially lead to dramatic advances in functionalization, not only as a structural template for inorganic compounds but also as the appearance of photoluminescence in systems incorporating metal complexes undergoing metallophilic interactions. Such functionalization could enable the dynamic structural transformations that lie at the very heart of bottom-up nanotechnology [58,63,64,65,66,67,68,69,70].

The work reported herein focused on the dynamic structural transformation of [Au(CN)_2_]^−^, [Ag(CN)_2_]^−^, and [Pt(CN)_4_]^2^^−^ through the use of diblock copolypeptide amphiphiles having the general structural formula poly-(L-lysine)-*block*-(L-cysteine) (Lys_m_-*b*-Cys_n_) (Figure 1). This study investigated the morphological evolutions associated with the metallophilic and polymeric interactions of [Au(CN)_2_]^−^, [Ag(CN)_2_]^−^, and [Pt(CN)_4_]^2^^−^ together with the hierarchical assembly of hybrid materials composed of combinations of the copolypeptides with the cyanometallates. The nature of the systematic assembly of these materials in solution was evaluated based on the results of spectroscopic and microscopic measurements.

## 2. Results and Discussion

The diblock copolypeptide amphiphiles Lys_390_-*b*-Cys_4_ (**1**), Lys_218_-*b*-Cys_4_ (**2**), and Lys_206_-*b*-Cys_4_ (**3**) were synthesized from amino acid *N*-carboxyanhydrides (NCAs) of *N*_ε_-benzyloxycarbonyl-L-lysine (*N*_ε_-Cbz-L-Lys) and *S*-carbobenzoxy-L-cysteine (*S*-Cbz-L-Cys), by ring-opening living polymerization using organometallic initiators as described in the Experimental section. The chain lengths for the Lys segments were determined using GPC, and the polydispersities (M_w_/M_n_) for the samples were found to range from 1.10 to 1.41, when polymerization of poly-(Cbz-Lys)_m_ was characterized. After determining the length of poly-(Cbz-Lys)_m_, the degree of copolymerization of the Cys portion in each specimen was calculated from the sulfur atom concentration, which was determined using inductively coupled plasma—optical emission spectroscopy. ^1^H NMR analyses in deuterium oxide indicated over 99.9% removal of the benzyloxycarbonyl groups from the Lys residues. Each of these diblock copolypeptide amphiphiles was subsequently used to prepare an aqueous solution. Similarly, aqueous solutions of K[Au(CN)_2_], K[Ag(CN)_2_l, and K_2_[Pt(CN)_4_] were prepared. These copolymer and metal complex solutions were then combined at a 1:1 mass ratio, resulting in [Au(CN)_2_]^−^, [Ag(CN)_2_l^−^, and [Pt(CN)_4_]^2^^−^ concentrations of 3.5, 5.0, and 2.7 mM, respectively. The calculated molar ratio between these potassium cyanometallate complexes and the Lys units in the diblock copolypeptide amphiphiles ranged from 0.56:1.0 to 1.0:1.0 (Table 1).

Scanning electron microscopy (SEM) was used to evaluate the morphological changes of these specimens on the mesoscopic scale to confirm the formation of nanohybrid assemblies. SEM observations of hybrids obtained by combining diblock copolypeptide amphiphiles **1**–**3** with [Au(CN)_2_]^−^ showed many particle-like structures approximately 500 nm in size (Figure 2) along with a number of rod-like structures with lengths of approximately 1 µm (Figure 2a). The structures of the hybrids obtained by combining the polypeptides with [Au(CN)_2_]^−^ were controlled by the assembly of each copolymer with the metal complex. In contrast, the **1**,**2**/[Ag(CN)_2_]^−^ hybrid was composed of multiple particulate morphologies arranged in clusters along with some dendritic structures (Figure 3a,b). Interestingly, the SEM images indicated that the **1**/[Ag(CN)_2_]^−^ hybrid had amorphous and indefinite structures attached to the surfaces of much more massive structures several tens of micrometers in size (Figure 3a). In the case of the **3**/[Ag(CN)_2_]^−^ hybrid, particle-like structures were attached to the surfaces of larger structures several micrometers in size (Figure 3c). The results of the difference between **1**–**3**/[Ag(CN)_2_]^−^ suggest that the [Ag(CN)_2_]^−^ with a smaller cyanometallate with Ag rather than cyanometallates with Au or Pt has a stronger electrostatic interaction with the Lys unit and a larger aggregation to microstructure. SEM observations of specimens made by combining copolymers **1** to **3** with [Pt(CN)_4_]^2^^−^ demonstrated the presence of numerous particle-like structures approximately 500 µm in size (Figure 4). Thus, the structural changes were dependent on cyanometallates with various metal species.

The detailed morphologies and diverse structures of the polypeptide/cyanometallate complex hybrids were also observed using high-angle annular dark-field scanning TEM (HAADF-STEM) (Figure 5). High-resolution STEM coupled with energy-dispersive X-ray spectroscopy (HR-STEM EDX) also confirmed that the hybrids consisted of both cyanometallate complexes and polypeptides. Figure 5 presents STEM-EDX maps of hybrids containing **1** with [Au(CN)_2_]^−^ (Figure 5a), [Pt(CN)_4_]^2^^−^ (Figure 5b), and [Ag(CN)_2_]^−^ (Figure 5c).

The size distributions of the nanostructures in water were also analyzed using dynamic light scattering at 25 °C (Figure 6 and Table 2) The data show multidisperse scattering with at least one or two peaks for each sample in the volume-based size distributions over the range of 50 to 1000 nm, in agreement with the SEM and TEM images (Figure 2, Figure 3, Figure 4 and Figure 5). In addition, the peaks had similar sizes although the copolymer lengths and Lys/Cys ratios were different between the samples. All the hybrids made with the Au complex showed peaks at 30–1200 nm, whereas they were at 100–400 nm for all hybrids with the Ag complex, and specimens incorporating the Pt complexes had peaks close to 200–350 nm. The presence of nanostructures several hundred nanometers in size was observed for all the hybrids, suggesting that particulate and aggregated structures, such as those seen in the SEM and TEM images, were also present in water.

The metallophilic interactions and self-assembly of the cyanometallate complexes with the polypeptides were also investigated by UV-visible absorption spectroscopy (Figure 7). The mixing of aqueous solutions of the various cyanometallate complexes with the polypeptides resulted in the appearance of shoulder peaks at approximately 250–320 nm. In the case of the Au complex hybrids, a shoulder appeared at 250–280 nm that was not produced by the pure complex in solution. This absorption (5*d*σ*→6*p*σ, Figure 8a) originated from Au-Au interactions [61,62]. The Ag complex hybrid also generated a shoulder at 230–250 nm that was also not seen in the spectrum of the pure complex and was also ascribed to an Ag-Ag interaction (4*d*σ*→5*p*σ, Figure 8b) [60], as in the case of the Au hybrid. The absorption peaks for the [Pt(CN)_4_]^2^^−^ hybrids at 200–270 nm were attributed to charge-transfer absorption for the Pt complex (5*d*→6*p*) or to interaction with the π-orbitals of the cyanide ligands [60]. Thus, these results confirm that metallophilic interactions took place, although these interactions did not greatly change the spectra because the associated electronic transitions are forbidden. In addition, it is noteworthy that the structural changes in the hybrids were dependent on the metal complex that was used, especially in the case of the Au complex.

The emission spectra of the mixtures of the cyanometallate complexes with the polypeptides were also investigated to assess the aggregation resulting from metallophilic interactions. The emission spectra of each hybrid are presented in Figure 9. A 1 mM cyanometallate concentration in the absence of a polypeptide did not exhibit luminescence. Luminescence was observed following the addition of the polypeptides, with an emission maximum at approximately 460 nm due to Au-Au interactions between the polynuclear [Au(CN)_2_]^−^_n_ excimer and exciplexes resulting from oligomers [61,62]. Luminescence with a shoulder at 350–420 nm was also observed, which indicated the formation of trimers or tetramers of [Au(CN)_2_]^−^. The red-shifted emission band at 460 nm suggests the formation of longer polynuclear Au complexes than those that were present in the initial solution of the Au complex. In the case of hybrids with the Ag complex, the emission peak was observed at approximately 410 nm and was attributed to interactions between [Ag(CN)_2_]^−^ dimers. The Pt complexes also showed an emission peak at approximately 420 nm, which was ascribed to interactions of Pt tetramers or disordered polynuclear complexes [60]. Thus, the emission wavelength differed depending on the degree of interaction between the metal complexes and the ability of the polypeptide to accumulate the metal complexes. Although no clear dependence in luminescence behavior or self-assembly behavior was observed depending on the length of the Lys segment, we were able to show that an appropriate length of polypeptide, around 200–400 mers, results in longer polynuclear Au complexes, which is consistent with the results in previous reports. [58] In addition, the supramolecular chemistry of Lys_m_-*b*-Cys_4_ in our own studies appears to have been affected by linear polynuclear Au-Au interactions rather than by the Ag-Ag or Pt-Pt interaction. In our manuscript, we also describe the effect of the polypeptide’s structure on Au-Au interactions and the luminescence in solution.

The hybrids made with the cyanometallate complexes showed UV absorption around 250–320 nm and emission around 400–500 nm, which was related to interactions between the metal complexes, specifically M-M-M-M multinuclear interactions. These results demonstrate that nanostructures were produced in all the hybrids, and the luminescence behavior was dependent on the integration state of the metal complexes. Since the g ratio for the hybrids was 1:1, the amount of each metal complex was small compared with the number of cationic Lys sites. The interactions between the metal complexes were evidently modified by generating the various hybrids based on electrostatic interactions between the cationic Lys moieties and the anionic complexes. This effect produced luminescence with higher quantum efficiency.

Circular dichroism (CD) spectra were obtained to determine the conformations of the polypeptides when combined with the metal complexes in water (Figure 10). In the case of the hybrids with Au and Ag complexes, a negative Cotton effect resulted from the formation of β-sheet and/or random coil structures (Figure 10a,b) [12,13,14,15,16,17,18]. In contrast, the Pt complex hybrids showed positive and negative Cotton effects with positive peaks at 195 nm and negative peaks at 205 and 222 nm, and a θ_222_/θ_205_ ratio of 0.47 (Figure 10c). These results are attributed to the formation of 3_10_-helix structures [71], including β-turn conformations [19]. It appears that [Pt(CN)_4_]^2^^−^_n_ units were assembled around segments of the amphiphile helical polypeptide backbone with the concurrent adoption of specific polypeptide conformations.

Fourier transform infrared (FTIR) spectra of these hybrids (Figure 11) were consistent with helical, sheet, and random coil structures, including the P^II^ (polyproline II-type) conformations observed in the CD spectra. A quantitative analysis of the amide bond I region (1600–1700 cm^−1^) provided information regarding changes in the secondary structures of the hybrids [72,73,74,75,76] (Figure 10 and Table 3). Interestingly, in the presence of [Au(CN)_2_]^−^ and [Ag(CN)_2_]^−^, contributions from the random coil and β-sheet conformations were observed. As an example, a β-sheet contribution of 32.0%, a random coil contribution of 57.4%, and a 3_10_-Helix or β-turn [75,76] contribution of 10.6% were found in the case of the **1**/[Au(CN)_2_]^−^ specimen. In contrast, in the presence of [Pt(CN)_4_]^2−^, the contribution from random coil conformations decreased. This resulted in a β-sheet contribution of 36.6 %, a random coil contribution of 46.7%, and a 3_10_-Helix or β-turn contribution of 10.6% for **1**/[Pt(CN)_4_]^2−^. The FTIR spectra of these hybrids showed various secondary structural components, such as β-sheet, 3_10_-Helix, or β-turn conformations, along with transitions from random coil to sheet or helix morphologies depending on the charge on the metal complex. Polypeptide amphiphiles in an aqueous solution can adopt α-helix or β-sheet conformations in the Lys segments. Therefore, the present results provide evidence for electrostatic interactions between the polypeptide segments and cyanometallate complexes, leading to sheet and helical backbone structures with random coil conformations.

The results of these morphological and spectroscopic investigations provide detailed information regarding the nature of the hybrids self-assembled from diblock copolypeptide amphiphiles and cyanometallate complexes. The observations of metallophilic interactions indicate that complex anions were assembled when combined with the polypeptides. SEM images confirmed the generation of nanoparticles, depending on the length and ratio of the hydrophilic/hydrophobic parts of the polypeptides. The UV-visible and emission spectra demonstrated that the hybrids included polynuclear species that underwent M-M bonding interactions. Electrostatic interactions between the amine segments and anionic metal complexes, as well as the resulting nanostructures, played an important role in enabling the metallophilic interactions. It is therefore evident that the polypeptides were capable of inducing detailed nanostructures based on helical and sheet conformations in aqueous solutions of metal complexes. In this manner, novel luminescent nanomaterials with tunable structures and luminescence behavior could be obtained.

## 3. Materials and Methods

### 3.1. Materials and Instrumentation

Tetrahydrofuran (THF) and hexane were dried by purging with nitrogen using a solvent purification apparatus (GlassContour, Nico-Hansen, Osaka, Japan). Co(PMe_3_)_4_ was prepared according to procedures previously published in the literature [57]. All chemicals were purchased from commercial suppliers (Tokyo Chemical Industry Co., Ltd., Tokyo, Japan; Fujifilm Wako Pure Chemical Co., Tokyo, Japan; Kanto Chemical Co., Inc., Tokyo, Japan; Sigma-Aldrich Chemical Co., St. Louis, MO, USA; Merck KGaA, Darmstadt, Germany) and used without further purification unless otherwise noted. Fourier transform infrared spectroscopy (FTIR) was carried out using a Spectrum 65 spectrometer (PerkinElmer, Inc., Waltham, MA, USA). Based on previously reported procedures [72,73,74], the secondary structures of the various samples and the number of residues therein were estimated. The FTIR spectra that were acquired suggested that the peptides included several structures, such as β-sheet, β-turn, 3_10_-helix, antiparallel β-sheet, and random coil conformations. [72,73,74,75,76] The peaks in the amide I region of each spectrum (1600–1700 cm^−1^) were assessed using a multiple Gaussian fitting procedure [75,76], and the proportion of each secondary structural constituent in the material was calculated using the Igor 9.0 software package. ^1^H nuclear magnetic resonance (NMR) spectra were acquired using an ESC 400 instrument (JEOL Ltd., Tokyo, Japan). Gel permeation chromatography/light scattering (GPC) was performed at 60 °C using a Shimadzu LC solution GPC system incorporating an RID-10A differential refractive index detector and a CBA-20A pump/controller (Shimadzu Co., Ltd., Kyoto, Japan). Separations were achieved using 10^5^, 10^3^, and 500 Å Phenomenex Phenogel 5 μm columns with 0.1 M LiBr in dimethylformamide as the eluent and sample concentrations of 1 mg/mL. Pyrogen-free deionized water was obtained from Direct-Q3-UV (Merck KGaA, Darmstadt, Germany) purification units. UV-visible and fluorescence spectra were obtained using RF-2500PC and RF-5300PC spectrophotometers, respectively (Shimadzu Co., Ltd., Kyoto, Japan). Circular dichroism spectra were acquired using a J-820 spectrophotometer (JASCO Corp., Tokyo, Japan). Scanning electron microscopy (SEM) was carried out with an ERA-600 microscopy (Elionix Inc., Tokyo, Japan) operating at 20 kV. SEM samples were prepared by transferring the surface layers of dispersions to Cu plates (Okenshoji Co., Ltd., Tokyo, Japan). Transmission electron microscopy (TEM) was performed using a Titan Themis 200 (Thermo Fisher Scientific Co. Ltd., Waltham, MA, USA) operating at 200 kV. TEM samples were prepared by transferring the surface layers of gels or solutions to carbon-coated grids (Okenshoji Co., Ltd., Tokyo, Japan). Inductively coupled plasma optical emission spectroscopy (ICP-OES) data were obtained with an iCAP 7400 instrument (Thermo Fisher Scientific Co. Ltd., Waltham, MA, USA).

### 3.2. General Polypeptide Synthesis

All diblock copolypeptide amphiphiles were synthesized using Co(PMe_3_)_4_ as the initiator and following a literature procedure (Figure 12) [57]. In each case, a 50 mL glass vial was charged with 0.5 g of the NCA of *N*_ε_-benzyloxycarbonyl-L-lysine (*N*_ε_-Cbz-L-Lys) and 5 mL of THF, after which this mixture was stirred in a glove box. The necessary amount of Co(PMe_3_)_4_ was then transferred to a 20 mL glass vial. 2.0 mL of the Co(PMe_3_)_4_/THF solution was transferred to the NCA/THF solution (50 mL vial) by syringe and stirred for 2 h. The amount of Co(PMe_3_)_4_ was based on a ratio of total moles of NCA monomer (L-Lys + L-Cys) to moles of Co(PMe_3_)_4_ equal to a quarter of polymerization degree. [39,40] Following this, the product was characterized by FTIR spectroscopy and GPC (Table 4). The contents of the 50 mL glass vial were injected into the 20 mL glass vial, which was then charged with the NCA of *S*-carbobenzoxy-L-cysteine (*S*-Cbz-L-Cys) and 5 mL of dry THF and stirred overnight. The resulting product was *N*_ε_-Cbz-L-Lys-*block*-*S*-Cbz-L-Cys. This material was also analyzed using FTIR spectroscopy. The *N*_ε_-Cbz-L-Lys-*block*-*S*-Cbz-L-Cys was subsequently transferred to a 100 mL flask and evaporated under vacuum. Following this, 40 mL of trifluoroacetic acid (TFA) was added, together with 4.4 mL of 33 wt% HBr in acetic acid, and the mixture was stirred for 1 h. The solid phase was removed and washed with diethyl ether and then dispersed in 30 mL of 0.1 M LiBr aqueous solution, and the sediment was transferred to a dialysis tube, after which a dialysis procedure was carried out for one week. During the first two days of this process, the tube was placed in 2.0 L of a 0.10 M aqueous EDTA solution that was replaced daily. Over the next three days, the tube was placed in 2.0 L of a 0.10 M aqueous LiBr solution, with daily replacement of the solution. Finally, during the last two days, the tube was placed in 2.0 L of deionized water, with daily replacement of the deionized water, and the sediment in the tube changed to a transparent solution. The dialyzed solution was then transferred to a centrifuge tube and freeze-dried to yield 50 mg of a colorless powder. The proportions of Lys and Cys units in this material were determined using ^1^H NMR spectroscopy with a 400 MHz instrument (Appendix A) and FT-IR spectroscopy (Appendix A). The degree of polymerization of the Cys portion was calculated from the sulfur atom content determined using ICP-OES (Table 4).

### 3.3. General Preparation of Polypeptide/Cyanometallate Complex Hybrids

Deionized water purged with nitrogen was used in these trials because of easy oxidative decomposition of cyanometallates. In each case, a quantity of the copolypeptide amphiphile (4 mg) was dissolved in deionized water (2 mL) and a portion of a potassium cyanometallate complex (4 mg) was dissolved in deionized water (2 mL). Hybrids were prepared by mixing both solutions to give combined polypeptide and complex with a nominal 1/1 ratio (g/g) in 4 mL water. The actual ratios were determined by ICP-OES analysis. The observed stabilization induced by adding the amphiphiles indicates that aggregation of the hybrids subsequent to the copolypeptides addition prevents cyanometallate complexes from reacting with oxygen at least for a month.

## 4. Conclusions

This work demonstrated the formation of hybrids composed of diblock copolypeptide amphiphiles with cyanometallates that had significant variations in nanostructure depending on the structure of the copolypeptide. The present results confirm that it is possible to control the metal-metal interactions of the complexes and to produce nanostructures based on aggregates. These supramolecular hybrids allow the design of flexible, reversible, and signal-responsive systems, and this general concept could be expanded to include other useful compounds. This research provides valuable information that is expected to lead to further advances in the fields of metalloproteins and biopolymer nanochemistry.

## Figures and Tables

**Figure 1 molecules-27-03262-f001:**
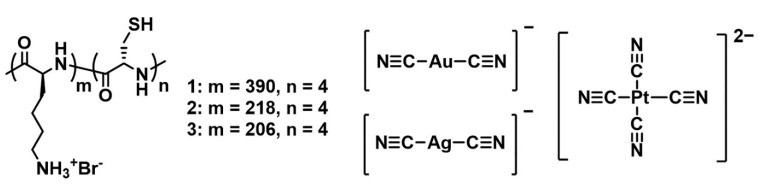
Molecular structures of the diblock copolypeptide amphiphiles Lys_390_-*b*-Cys_4_ (**1**), Lys_218_-*b*-Cys_4_ (**2**), and Lys_206_-*b*-Cys_4_ (**3**), and of the cyanometallate complexes [Au(CN)_2_]^−^, [Ag(CN)_2_]^−^, and [Pt(CN)_4_]^2^^−^.

**Figure 2 molecules-27-03262-f002:**

SEM images of (**a**) **1**/[Au(CN)_2_]^−^, (**b**) **2**/[Au(CN)_2_]^−^, and (**c**) **3**/[Au(CN)_2_]^−^, where [Au(CN)_2_]^−^ = 3.5 mM.

**Figure 3 molecules-27-03262-f003:**
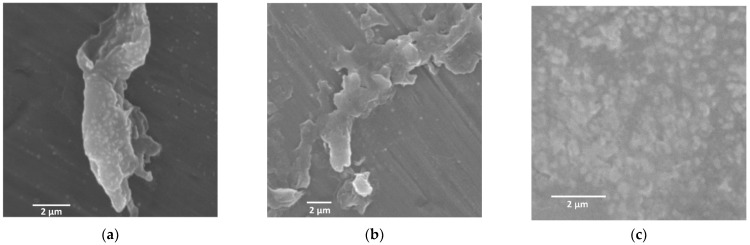
SEM images of (**a**) **1**/[Ag(CN)_2_]^−^, (**b**) **2**/[Ag(CN)_2_]^−^, and (**c**) **3**/[Ag(CN)_2_]^−^, where [Ag(CN)_2_]^−^ = 5.0 mM.

**Figure 4 molecules-27-03262-f004:**
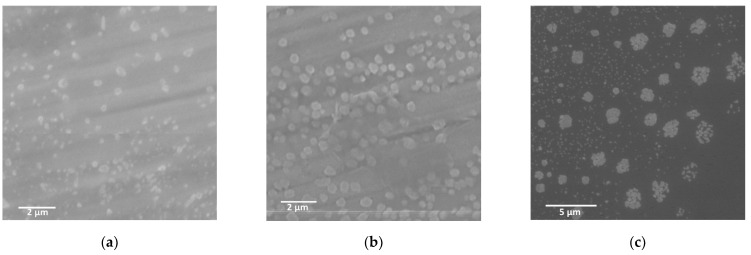
SEM images of (**a**) **1**/[Pt(CN)_4_]^2−^, (**b**) **2**/[Pt(CN)_2_]^2−^, and (**c**) **3**/[Pt(CN)_4_]^2−^, where [Pt(CN)_4_]^2−^ = 2.7 mM.

**Figure 5 molecules-27-03262-f005:**
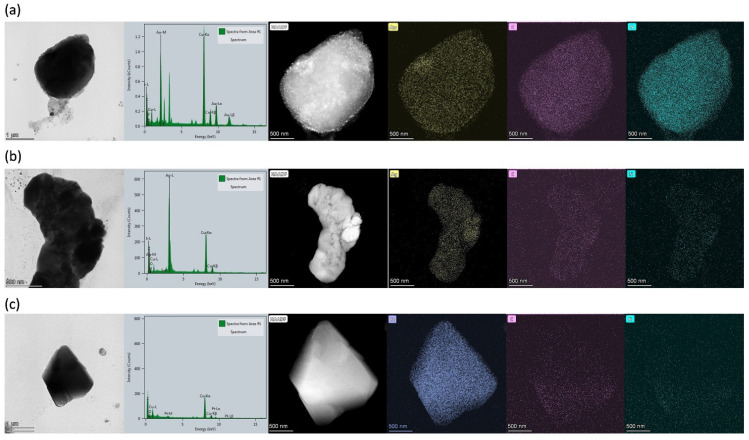
HAADF-STEM images and STEM-EDX maps (showing metals (Au, Ag, Pt), C, and N) for hybrids of (**a**) [Au(CN)_2_]^−^, (**b**) [Ag(CN)_2_]^−^, and (**c**) [Pt(CN)_4_]^2−^ with diblock copolypeptide amphiphile **1**, where [Au(CN)_2_]^−^ = 3.5 mM, [Ag(CN)_2_]^−^ = 5.0 mM, and [Pt(CN)_4_]^2−^ = 2.7 mM.

**Figure 6 molecules-27-03262-f006:**
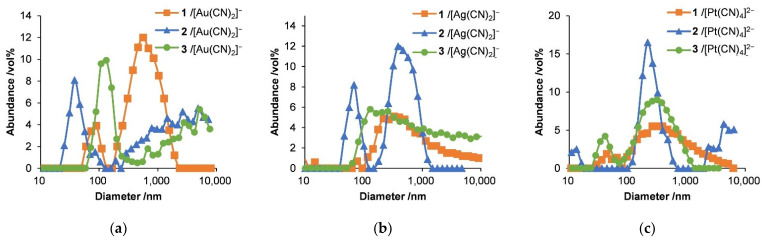
Dynamic light scattering results for hybrids of (**a**) [Au(CN)_2_]^−^, (**b**) [Ag(CN)_2_]^−^, and (**c**) [Pt(CN)_4_]^2−^ with diblock copolypeptide amphiphiles 1–3, where [Au(CN)_2_]^−^ = 3.5 mM, [Ag(CN)_2_]^−^ = 5.0 mM, and [Pt(CN)_4_]^2−^ = 2.7 mM.

**Figure 7 molecules-27-03262-f007:**
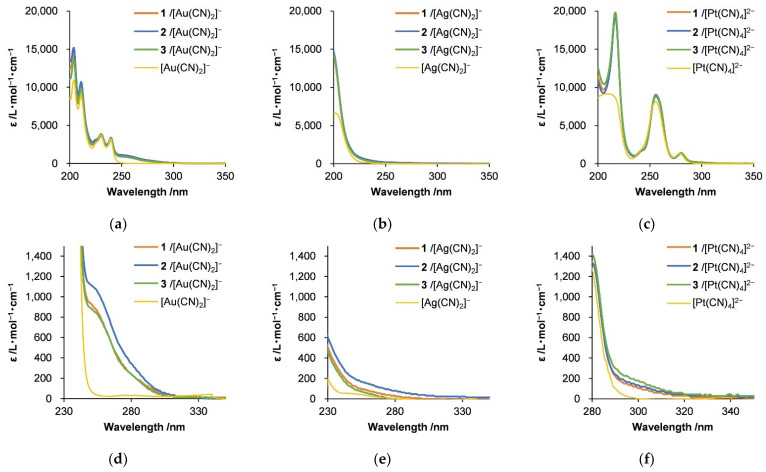
UV-visible spectra of hybrids of (**a**) [Au(CN)_2_]^−^, (**b**) [Ag(CN)_2_]^−^, and (**c**) [Pt(CN)_4_]^2−^ with diblock copolypeptide amphiphiles 1–3, where [Au(CN)_2_]^−^ = 3.5 mM, [Ag(CN)_2_]^−^ = 5.0 mM, and [Pt(CN)_4_]^2−^ = 2.7 mM. (**d**–**f**) Spectra enlarged to show region from 230 to 350 nm to focus on metal-metal interactions.

**Figure 8 molecules-27-03262-f008:**
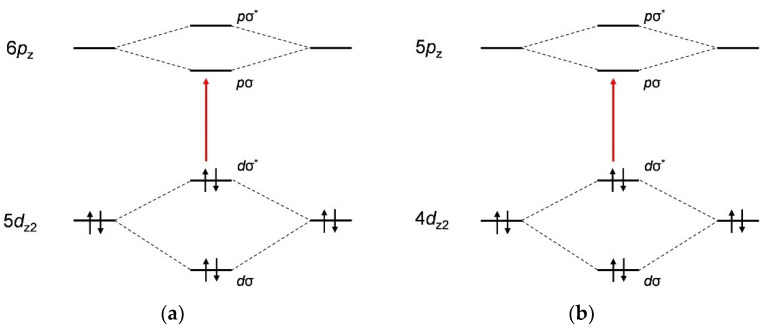
Molecular orbital diagrams for (**a**) Au(I), Pt(II) and (**b**) Ag(I) in cyanometallate complexes showing effective metal-metal electronic interactions based on z-axis stacking.

**Figure 9 molecules-27-03262-f009:**
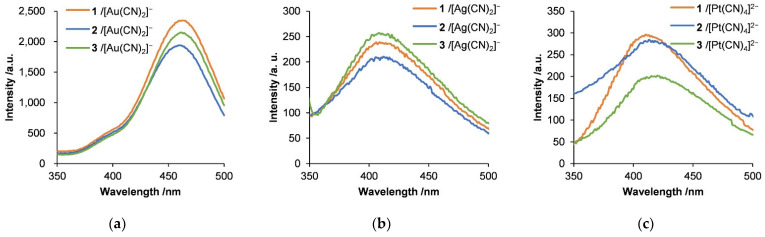
Emission spectra of hybrids of (**a**) [Au(CN)_2_]^−^, (**b**) [Ag(CN)_2_]^−^, and (**c**) [Pt(CN)_4_]^2−^ with diblock copolypeptide amphiphiles 1–3, where [Au(CN)_2_]^−^ = 3.5 mM, [Ag(CN)_2_]^−^ = 5.0 mM, and [Pt(CN)_4_]^2−^ = 2.7 mM. Excitation wavelength = 335 nm.

**Figure 10 molecules-27-03262-f010:**
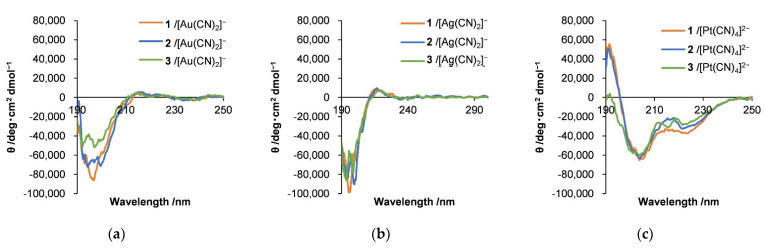
Circular dichroism spectra obtained from hybrids of (**a**) [Au(CN)_2_]^−^, (**b**) [Ag(CN)_2_]^−^, and (**c**) [Pt(CN)_4_]^2−^ with diblock copolypeptide amphiphiles 1–3, where [Au(CN)_2_]^−^ = 3.5 mM, [Ag(CN)_2_]^−^ = 5.0 mM, and [Pt(CN)_4_]^2−^ = 2.7 mM.

**Figure 11 molecules-27-03262-f011:**
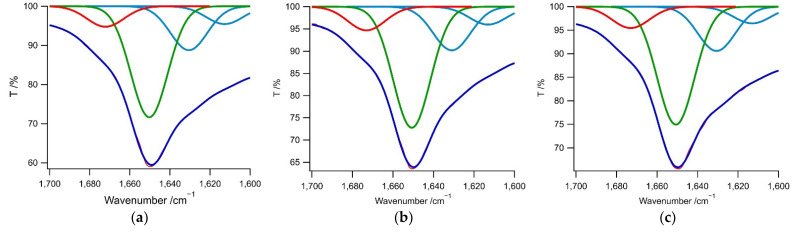
FTIR spectra (purple curves) in the amide I region obtained from the (**a**–**c**) **1**–**3**/[Au(CN)_2_]^−^, (**d**–**f**) **1**–**3**/[Ag(CN)_2_]^−^, and (**g**–**i**) **1**–**3**/[Pt(CN)_4_]^2−^ hybrids, where [Au(CN)_2_]^−^ = 3.5 mM, [Ag(CN)_2_]^−^ = 5.0 mM, and [Pt(CN)_4_]^2^^−^ = 2.7 mM. Multiple Gaussian fitting was used to determine the secondary structural components. Blue, green, red, and orange curves denote the amide I features related to β-sheet (1610–1640 cm^−1^), random coil (1640–1660 cm^−1^, including P^II^ structures), 3_10_-helix or β-turn-like (1660–1685 cm^−1^), and antiparallel β-sheet structures (1675–1690 cm^−1^), respectively [72,73,74,75,76]. The overall curve fittings are indicated by the red lines.

**Figure 12 molecules-27-03262-f012:**
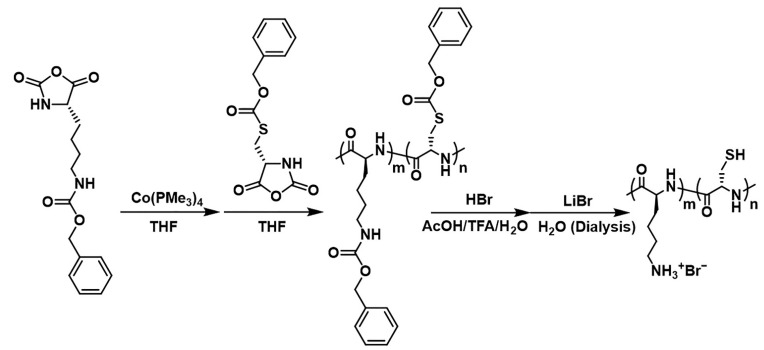
Synthetic scheme of diblock copolypeptide amphiphiles 1–3.

**Table 1 molecules-27-03262-t001:** Calculated concentrations of potassium cyanometallate complexes and of repeating units in diblock copolypeptide amphiphiles in aqueous solutions prepared at a 1:1 mass ratio (that is: 1 mg to 1 mg in 1 mL of water).

	Cyanometallic Complexes	Lys	Cys
	/mM	/mM	/mM
1/[Au(CN)_2_]^−^	3.5	4.8	0.049
2/[Au(CN)_2_]^−^	3.5	4.8	0.088
3/[Au(CN)_2_]^−^	3.5	4.8	0.093
1/[Ag(CN)_2_]^−^	5.0	4.8	0.049
2/[Ag(CN)_2_]^−^	5.0	4.8	0.088
3/[Ag(CN)_2_]^−^	5.0	4.8	0.093
1/[Pt(CN)_4_]^2−^	2.7	4.8	0.049
2/[Pt(CN)_4_]^2−^	2.7	4.8	0.088
3/[Pt(CN)_4_]^2−^	2.7	4.8	0.093

**Table 2 molecules-27-03262-t002:** Maximum values in the volume-based size distributions presented in dynamic light scattering results.

	Maximum Values
	/nm
1/[Au(CN)_2_]^−^	460
2/[Au(CN)_2_]^−^	38
3/[Au(CN)_2_]^−^	1139
1/[Ag(CN)_2_]^−^	332
2/[Ag(CN)_2_]^−^	389
3/[Ag(CN)_2_]^−^	130
1/[Pt(CN)_4_]^2−^	317
2/[Pt(CN)_4_]^2−^	216
3/[Pt(CN)_4_]^2−^	314

**Table 3 molecules-27-03262-t003:** Data obtained from amide peaks in FTIR spectra indicating the percentage contributions of β-sheet (1610–1640 cm^−1^), random coil (1640–1660 cm^−1^, including P^II^ structures), 3_10_-helix or β-turn-like (1660–1685 cm^−1^), and antiparallel β-sheet (1675–1690 cm^−1^) secondary structures [72,73,74,75,76] for hybrid **1**–**3**/cyanometallate complexes.

Pattern of Secondary Structure	β-Sheet		Random Coil		3_10_-Helix, β-Turn		Antiparallel β-Sheet	
	Peak (cm^−1^)	Percent	Peak (cm^−1^)	Percent	Peak (cm^−1^)	Percent	Peak (cm^−1^)	Percent
**1**/[Au(CN)_2_]^−^	1612, 1630	32.0	1650	57.4	1672	10.6		
**2**/[Au(CN)_2_]^−^	1613, 1631	29.8	1651	58.7	1673	11.5		
**3**/[Au(CN)_2_]^−^	1613, 1630	23.0	1651	58.9	1673	10.7		
**1**/[Ag(CN)_2_]^−^	1613, 1631	25.2	1651	62.5	1672	12.3		
**2**/[Ag(CN)_2_]^−^	1630	20.1	1651	66.7	1672	13.2		
**3**/[Ag(CN)_2_]^−^	1630	18.9	1651	67.2	1672	13.9		
**1**/[Pt(CN)_4_]^2−^	1616, 1633	36.6	1650	46.7	1667	10.6	1679	6.0
**2**/[Pt(CN)_4_]^2−^	1615, 1632	35.2	1650	47.2	1666	10.2	1679	6.8
**3**/[Pt(CN)_4_]^2−^	1618, 1637	38.5	1652	47.1	1672	12.1	1686	2.3

**Table 4 molecules-27-03262-t004:** Properties of the diblock copolypeptide amphiphiles synthesized in this work.

Predicted Composition	M_n_ ^a^	M_w_/M_n_ ^a,b^	Lys_m_ Length ^c^	Found Composition ^d^
(Lys)_390_-*block*-(Cys)_4_	9.52X10^4^	1.41	(Lys)_390_	(Lys)_390_-*block*-(Cys)_4_
(Lys)_218_-*block*-(Cys)_4_	5.38X10^4^	1.14	(Lys)_218_	(Lys)_218_-*block*-(Cys)_4_
(Lys)_206_-*block*-(Cys)_4_	5.04X10^4^	1.10	(Lys)_206_	(Lys)_206_-*block*-(Cys)_4_

^a^ Determined using gel permeation chromatography based on *N*_ε_-Cbz-Lys units. ^b^ M_w_/M_n_ = Polydispersity index. ^c^ Determined from M_n_ measurements. ^d^ Determined from M_n_ measurements and ^1^H NMR and ICP analysis of deprotected samples.

## Data Availability

Not applicable.

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
