# Peer review of "Supramolecular Hybrids from Cyanometallate Complexes and Diblock Copolypeptide Amphiphiles in Water"

_molecules, 2022, doi:10.3390/molecules27103262_

Round 1

Reviewer 1 Report

The manuscript “Supramolecular Hybrids from Cyanometallate Complexes and Diblock Copolypeptide Amphiphiles in Water” by Takayuki Tanaka , Keita Kuroiwa concerns the  formation of hybrids composed of diblock copolypeptide amphiphiles with cyanometallates having significant variations in nanostructure depending on the construction of the copolypeptide. The article is very relevant and interesting. The studies were performed using the most modern research methods. However, significant shortcomings in the experimental part and in "Results and discussions" require a significant revision of the manuscript.

"Molecules" is a scientific journal of a wide profile and is aimed at a non-specialized readership. Therefore, the authors should consider this and introduce the reader to the essence of the issue under study in more detail. From this point of view, the introduction is very poorly written. Highlighted in green words or expressions (see the pdf-file) are unsuccessfully used, making it difficult to understand the text.

The main disadvantage of section 3 is that neither the manuscript nor the cited article indicates the specific amount of the initiator, which does not allow one to reproduce the synthesis of the diblock copolymer.

The authors also did not indicate in the form of which salts (cations) the complex anions were used.

To improve the perception of the text, additional figures and tables should be provided.

For example:

Add a scheme with general chemical structure of NCA and ring‐opening living polymerization

Please list in tabular form the maximum values of the peaks presented in figure 6.

Line 60. What das it mean “hierarchical” in this case? Explain for readers please.

Please provide the spectra (UV‐visible absorption spectroscopy) of aqueous solutions of pure cyanometallates.

What explains the different dependence of the intensity of the emission peak on the value of m in the diblock copolymers (Lys)m-b-Cys4 for different cyanometallates.

Add “cyanometallates” in keyword list.

As unpublished material, the authors attached a file of “a letter to the editor”.

None of the "live" NMR or IR spectra for supramolecular hybrids is presented in the publication.

A complete list of comments and commentaries is provided directly in the pdf file of the manuscript.

Reviewer 2 Report

This ms. by Tanaka and Kuroiwa describes the study of supramolecular hybrids containing cyanometallates with diblock copoly‐ 13 peptide amphiphiles. The structures have been studied in depth using a variety of techniques, including SEM, TEM, DLS, UV-Vis, CD and FT-IR. The ms. is written in a nice manner, well-organized and the results are supported by the scientific data. This work is of particular interest to the scientist working in the field of diblock copolypeptide amphiphiles and of supramolecular chemistry in general. I don’t have any comments/corrections and thus I gladly recommend its publication in Molecules in its present form.

Reviewer 3 Report

Comments to the Molecules 1711170. 

This paper was written pretty well. I have nothing to comment. 

However, I am not satisfied with the explanation of SEM figures, especially Fig 3. 

Reviewer 4 Report

The article 'Supramolecular Hybrids from Cyanometallate Complexes and Diblock Copolypeptide Amphiphiles in Water' describes supramolecular assembly of the discrete cyanometallates in the presence of poly(L-lysine)-b-poly(L-cysteine) in water.

The subject of the study and the results obtained are of interest for a readership of the Molecules journal. The quality of expreiment and presentation is high, the article can be published after minor additions and corrections.

line 17, 98 and below – replace 'poly-L-lysine-block-L-cysteine' by 'poly(L-lysine)-b-poly(L-cysteine)'

line 112 and below – L- (configuration of amino acids) should be italic (L-)

I also recommend to supplement Fig. 1 by the scheme of the synthesis of poly(L-lysine)-b-poly(L-cysteine).

Round 2

Reviewer 1 Report

The text of the manuscript, data, and illustrative material have been significantly improved by the authors, however, some of the corrections made require minor adjustments.
Please rephrase the sentence (lines 159-160). It turned out very sloppy. At the end of the sentence (line 166), leave only "on cyanometallate nature."
